# FS-RSDD: Few-Shot Rail Surface Defect Detection with Prototype Learning

**DOI:** 10.3390/s23187894

**Published:** 2023-09-15

**Authors:** Yongzhi Min, Ziwei Wang, Yang Liu, Zheng Wang

**Affiliations:** 1School of Automation and Electrical Engineering, Lanzhou Jiaotong University, Lanzhou 730070, China; 12211463@stu.lzjtu.edu.cn; 2School of Mechanical Engineering, Lanzhou Jiaotong University, Lanzhou 730070, China; wangz@lzjtu.edu.cn

**Keywords:** rail surface defect detection, few-shot learning, prototype learning, transfer learning, unsupervised anomaly detection

## Abstract

As an important component of the railway system, the surface damage that occurs on the rails due to daily operations can pose significant safety hazards. This paper proposes a simple yet effective rail surface defect detection model, FS-RSDD, for rail surface condition monitoring, which also aims to address the issue of insufficient defect samples faced by previous detection models. The model utilizes a pre-trained model to extract deep features of both normal rail samples and defect samples. Subsequently, an unsupervised learning method is employed to learn feature distributions and obtain a feature prototype memory bank. Using prototype learning techniques, FS-RSDD estimates the probability of a test sample belonging to a defect at each pixel based on the prototype memory bank. This approach overcomes the limitations of deep learning algorithms based on supervised learning techniques, which often suffer from insufficient training samples and low credibility in validation. FS-RSDD achieves high accuracy in defect detection and localization with only a small number of defect samples used for training. Surpassing benchmarked few-shot industrial defect detection algorithms, FS-RSDD achieves an ROC of 95.2% and 99.1% on RSDDS Type-I and Type-II rail defect data, respectively, and is on par with state-of-the-art unsupervised anomaly detection algorithms.

## 1. Introduction

The rapid growth of railway operation mileage in recent years, due to the construction of numerous new railway lines in many countries, has significantly increased the pressure on maintenance. During the daily operation of railway systems, the interaction between wheels and rails inevitably leads to surface defects such as spalling, corrugation, and grinding, which pose serious hidden dangers to safe operation. Unlike internal defects in rails that can be detected using techniques such as ultrasound [1] and eddy current [2,3], traditional rail surface defect detection is mainly conducted through manual visual inspection, which is inefficient and heavily relies on human workers’ experience [4]. In recent years, many researchers have focused on developing machine vision-based rail surface defect detection technologies that offer higher efficiency and accuracy to address the aforementioned issues. With the rapid development of artificial intelligence technology, deep-learning-based algorithms, specifically supervised learning-based defect detection algorithms, are being widely applied in rail surface defect detection [5,6,7,8].

However, defect samples are difficult to obtain in practical work; thus, defect detection methods based on supervised learning face two important challenges due to insufficient defect samples. One of them is the risk of overfitting caused by the limited training data, which may not adequately represent the distribution of defect; additionally, supervised learning methods typically require the use of a portion of the defect data for training, leading to a reduction in the number of testing samples available for validation, which affects the credibility of the validation results. Inspired by the concept of anomaly detection (AD), some researchers have turned their attention to utilizing unsupervised learning techniques to address the aforementioned issues in the field of defect detection [9,10,11]. However, these unsupervised learning-based methods rely completely on modeling the distribution of normal samples, lacking an understanding of defect data, which may lead to poor classification performance and a potentially high false-positive/negative rate [12].

To tackle the aforementioned shortcomings of supervised learning-based defect detection methods, this paper proposes a few-shot rail surface defect detection model called FS-RSDD (few-shot rail surface defect detection). Inspired by the prototype learning and feature-embedding-based unsupervised AD (anomaly detection algorithms), FS-RSDD uses a pre-trained neural network as a feature extractor for both normal and defective rail images. Global average pooling and mask average pooling are used to embed features for normal and defective samples, respectively, which aim to compress the feature maps into feature vectors to obtain a compact feature memory bank. Subsequently, an unsupervised learning algorithm is used to obtain the feature prototypes of normal samples. Finally, the detection of rail defects is accomplished through the similarity computation between input features and prototypes. In summary, our main contributions are as follows:To overcome the challenges associated with using supervised learning-based defect detection algorithms when there is insufficient defect data available, we have introduced a simple yet effective few-shot rail surface defect detection method called FS-RSDD, which combines unsupervised anomaly detection with prototype learning. By effectively integrating the feature prototypes of normal rail images and defect rail images, we have achieved high accuracy in detecting rail surface defects with very little defect samples used for training.By avoiding the partitioning of normal rail backgrounds into small image patches and individually modeling the feature distribution of each image patch, FS-RSDD achieves a compact feature memory bank for normal rail samples, alleviating the issue of memory bank redundancy in feature-embedding-based unsupervised anomaly detection algorithms.FS-RSDD extensively leverages the fusion of multi-scale features to improve prediction accuracy. Furthermore, due to the integration of both normal background feature prototypes and defect feature prototypes for defect detection, the performance of the FS-RSDD model remains stable and robust compared to other few-shot industrial defect detection algorithms, even when the quality of the defect samples used for training is relatively low.Through extensive experiments, our method outperformed most existing few-shot supervised defect detection algorithms under the same number of defect samples used for training and achieved comparable performance to existing unsupervised anomaly detection algorithms which assume the availability of normal training samples only.

## 2. Related Works

### 2.1. Rail Surface Defect Detection

Previous research on rail surface defect detection often utilizes traditional image processing techniques to extract features from defect images and trains detection models using corresponding machine learning methods [13,14,15,16]. However, the performance of these methods is limited by the design of feature extraction, and the detection results can easily be affected by factors such as lighting, noise, and other factors. With the rapid development of deep learning technology, an increasing number of researchers have started studying rail surface defect detection methods based on deep learning, especially supervised learning methods. Wang Hao et al. integrated the improved pyramid feature fusion and modified loss function into the Mask-RCNN algorithm for the purpose of detecting rail surface defects [4]. Meng Si et al. proposed a multi-task architecture for rail surface defect detection, which includes two branch models for rail detection and defect segmentation [17]. Zhang Hui et al. cascaded the one-stage object detection algorithms SSD and YOLOv3, integrating the detection results from both networks to improve the accuracy of rail surface defect detection [18]. However, these approaches neglected the fact that defect samples are scarce and difficult to obtain in practical work.

Due to the limited number of defect samples in the field of defect detection, supervised algorithms-based defect detection models often face issues of overfitting and low validation credibility. To address these problems, many researchers have proposed corresponding solutions. D. Zhang et al. partitioned the rail image data into multiple segments and trained the defect detection model. However, this approach did not fundamentally solve the problem [19], and more researchers have recently started studying steel rail surface defect algorithms based on unsupervised anomaly detection algorithms. Q. Zhang et al. implemented the detection of rail surface defects using the multi-scale cross FastFlow model [20], while Menghui Niu et al. proposed an unsupervised stereoscopic saliency detection method for detecting rail surface defects and achieved good detection results [21]. However, some studies have pointed out that unsupervised anomaly defect detection algorithms often lead to a higher false detection rate [22,23] due to the lack of knowledge about defect samples during the training process. In this paper, we propose a simple yet effective few-shot rail surface defect detection algorithm that fully utilizes the feature information of normal steel rail samples and defect sample information to achieve defect detection.

### 2.2. Unsupervised Anomaly Detection for Industrial Images

Deep-learning-based algorithms are being widely used in industrial defect detection research in recent years due to their high efficiency and accuracy. Many researchers have devoted themselves to researching industrial defect detection algorithms based on supervised learning algorithms, which significantly depends on labeled defect data [24,25,26,27,28,29]. However, due to the hardship of collecting defective samples, it is extremely hard to obtain enough defect data for a deep model to learn its distribution. Furthermore, supervised learning-based methods require defect data for training, which further restricts the quantity of test datasets and affects the credibility of validation performance. In recent years, unsupervised-based anomaly detection (AD) algorithms have become the mainstream paradigm for industrial defect detection, which can be categorized as reconstruction-based and feature-embedding-based [30,31,32].

Reconstruction-based methods aim to train a deep network such as an adversarial generative network (GAN) or auto encoder (AE) to reconstruct normal images. When defective images are fed into the network, the defective parts cannot be reconstructed well, allowing for the detection of defects. However, sometimes the model can also yield a good reconstruction for the defective parts due to the powerful ability of the deep model [30].

Feature-embedding-based methods became the prevalent architecture in recent years, which typically consisted of a feature extractor and a feature estimator. A feature extractor is a deep network, typically a ResNet [33], that is pre-trained on ImageNet datasets. It is used to extract features from normal images, which are then stored into a memory bank. A feature estimator is used to estimate the distribution for normal features, which can be a multidimensional Gaussian distribution [34], clustering methods [35], or flow-based methods [36]. To avoid the deviation caused by different data distribution between industrial images and ImageNet datasets, only features from shallow layers are used. After distribution estimation, a distance metric is typically used to detect defects, since defects should be far from the center of the estimated distribution. One major drawback of embedding-based anomaly detection algorithms is that they estimate the distribution separately for each patch of the feature map, resulting in a massive and redundant feature memory bank to restore features from each patch. Many researchers have tried different methods to alleviate the problem: Padim experimentally studied the possibility to reduce redundancy of the memory bank and eventually chose to randomly discard a portion of the extracted features [30]; Patchcore utilized a coreset subsampling method to select representative features [32], thereby compressing the size of the feature memory bank. This paper introduces a feature representation method widely used in few-shot learning, which obtains a representative and compact feature memory bank and alleviates the aforementioned redundancy problem of the memory bank for rail surface defect detection.

### 2.3. Few-Shot Learning

In recent years, deep learning algorithms based on supervised learning have garnered significant attention from researchers due to the remarkable ability of deep models and large-scale datasets with high-quality labels. However, it is well known that supervised algorithms fail to acquire strong generalization ability when trained on a dataset with a small amount of data. Moreover, in many fields such as industrial defect detection, collecting a large-scale dataset with high-quality annotations proves to be challenging. This realization has prompted many researchers to shift their focus to the field of few-shot learning, with the aim of enabling the model to obtain strong generalization ability with only a few samples, akin to human beings.

Within the domain of few-shot learning in computer vision, image classification tasks are a prominent focal point. These tasks can be broadly categorized into three distinct classes: data-augmentation-based methods, parameter-optimization-based methods, and metric-learning-based methods.

Data-augmentation-based methods aim to address the challenge of limited samples in few-shot learning indirectly by enhancing the intricacy of the dataset through data augmentation. Trinet [37] employs autoencoders to map the features to the semantic space, followed by mapping the augmented features back to the sample space via semantic nearest neighbor search. Moreover, Patchmix [38] resolves the issue of distribution shift by substituting a specific region of the query image with random gallery images from diverse categories.

Parameter-optimization-based methods generally first train a meta-learner to learn common features (prior knowledge) of different tasks and then apply the obtained meta-knowledge to fine-tune the base learner on the query set. The model-agnostic meta-learning (MAML) [39], which first trains the model on a large number of task sets to obtain an adaptable weight and then fine-tunes the model on the target task to obtain the final classifier.

Metric-learning-based methods leverage pre-trained neural networks to extract features from training data. These extracted features are then utilized to measure similarity between the training data and test data using a metric. Representative methods include Siamese networks [40] and matching networks [41]. The former inputs two samples into the neural network and compares the similarity of the output feature vectors, while the latter uses attention mechanisms to obtain information about the correlation between feature vectors.

A typical embedding-based approach to few-shot image classification is the prototypical network [42], which utilizes a pre-trained model to extract features from a limited amount of labeled data and learns corresponding feature prototypes from them. The network then produces a distribution over classes for an input feature based on a softmax function over distances to the prototypes in the embedding space.

The prototypical network approach, combined with the utilization of mask average pooling, has been widely adopted in few-shot semantic segmentation methods. In addition, the idea of prototype features in prototypical networks has also been widely applied in many unsupervised anomaly detection algorithms [43,44].

## 3. Methods

This paper proposes an approach for rail surface defect detection called FS-RSDD. It aims to tackle the challenge of detecting surface defects with a limited number of defect samples. The proposed model combines defect feature prototypes and background feature prototypes to enable few-shot learning in this task. The architecture of the model is depicted in Figure 1, illustrating the integration of the proposed approach.

Figure 1 depicts the proposed method, which consists of two parts: embedding extraction and prediction. In the embedding extraction phase, the approach is inspired by feature-embedding-based anomaly detection techniques. A pre-trained model is employed for extracting multi-scale features from the training set images. These extracted features are then processed to generate a compact memory bank.

During the prediction phase, the feature prototypes obtained from the embedding extraction phase are utilized to calculate the multi-scale similarity feature maps with the feature map of test images. These similarity feature maps of normal and defect samples are then synthesized at each scale to generate a segmentation probability map. Finally, the probability map is smoothed to obtain the final prediction result. This process enables the detection of rail surface defects with high accuracy with limited defect samples.

### 3.1. Embedding Extraction

In this paper, a ResNet g(·) pre-trained on the public dataset ImageNet is employed as a feature extractor, and k is defined as a layer index of ResNet. In order to avoid the deviation caused by different data distribution between industrial images and ImageNet datasets, only features from first three layers are used; thus, k∈{1,2,3}.

First, the pre-trained model weights are fixed, and then the training set images are passed through the feature extractor. Next, the feature maps are extracted from the shallow layer of the network. Specifically, we are presented with $\left\{N_{train},D_{train}\right\}$, in which subset $N_{train}=\{x_{1},x_{2},\cdots ,x_{N}\}$ only contains normal samples and subset $D_{train}=\{x_{N+1},x_{N+2},\cdots ,x_{N+M}\}$ only contains defect samples with N≫M. As shown in Equations (1) and (2), $F_{D}^{k}$ and $F_{N}^{k}$ refer to the defect feature maps and normal feature maps, respectively. They are obtained from the *k*-th layer of the feature extractor, which is denoted as $g_{k}(\cdot )$. M and N refer to number of defect samples and normal samples respectively. $C_{k}$, $H_{k}$, $W_{k}$ refer to channels, height, and width of feature map from layer *k*.
(1)$$F_{N}^{k}=g_{k}\left(N_{train}\right),\;F_{N}^{k}\in R^{N\times C_{k}\times H_{k}\times W_{k}}$$
(2)$$F_{D}^{k}=g_{k}\left(N_{train}\right),\;F_{D}^{k}\in R^{M\times C_{k}\times H_{k}\times W_{k}}$$

After obtaining the feature representations from defective and normal rail images, the corresponding feature memory bank can be created by the proposed process.

### 3.2. Compact Multi-Scale Memory Bank

After obtaining the corresponding feature maps, the global average pooling (GAP) operation is applied to the feature maps of normal rail images. This operation fuses the global information of normal samples into a feature vector. On the other hand, for defective rail images, since the defective parts only occupy a small portion of the entire image, the mask average pooling (MAP) operation is used. This operation, as shown in Figure 2, is widely employed in few-shot semantic segmentation. It eliminates the features of normal parts in the feature map and only preserves the defect-specific features by element-wise production between feature map and mask, and then global average pooling is applied to obtain the prototype of defects.

In Equations (3) and (4), *GAP* represents the global average pooling operation, ${mask}_{Dj}^{k}$ represents the ground truth mask, $p_{D}^{k}(x_{j})$ represents the feature prototype, both ${mask}_{Dj}^{k}$ and $p_{D}^{k}(x_{j})$ correspond to a certain defective sample $x_{j}$, and $p_{n}^{k}(x_{i})$ represents the feature prototype of the normal sample $x_{i}$. Additionally, $f_{D}^{k}(x_{j})$ indicates a feature map corresponding to a certain image $x_{j}$, and $f_{N}^{k}(x_{i})$ indicates a feature map corresponding to $x_{i}$. ⊙ indicates the Hadamard product.
(3)$$p_{D}^{k}\left(x_{j}\right)=GAP\left(f_{D}^{k}\left(x_{j}\right)⊙{mask}_{Dj}^{k}\right),\;p_{D}^{k}\in R^{C_{k}},x_{j}\in D_{train}$$
(4)$$p_{N}^{k}\left(x_{i}\right)=GAP\left(f_{N}^{k}\left(x_{i}\right)\right),\;p_{N}^{k}\in R^{C_{k}},x_{i}\in D_{train}$$

The global average pooling operation is shown in Equation (5), where $p_{N}^{k}(x_{i})$ represents the normal feature prototype obtained by applying global average pooling to a certain normal feature in the layer *k*, and $f_{N}^{k}(x_{i})(h,w)$ represents the value of feature map $f_{N}^{k}(x_{i})$ at position (*h*,*w*).
(5)$$p_{N}^{k}\left(x_{i}\right)=\frac{1}{H_{k}\cdot W_{k}}\sum_{h=1}^{H_{k}}\sum_{w=1}^{W_{k}}f_{N}^{k}\left(x_{i}\right)\left(h,w\right),\;p_{N}^{k}\in R^{C_{k}}$$

As the number of normal samples used is significantly higher than the number of defect samples, which is distinct from the few-shot learning scenario, unsupervised algorithms can be used to obtain the distribution of normal sample features. Instead of estimating the feature prototype using the mean of sample features, as carried out in the prototypical network, this study adopts a widely used clustering algorithm, K-Means, to cluster the normal sample features. The cluster centers are then used as the final feature prototypes of the normal samples.

For the normal sample feature prototype, which consists of a set of feature vectors, clustering is performed with a predetermined number of clusters denoted as n. In this study, a cluster center number of 30 is chosen to cluster the normal samples, and the resulting cluster centers are utilized as the final feature prototypes. Since the number of defect sample features is relatively small, no clustering is conducted, and they are directly used as feature prototypes. All prototypes will be stored as a memory bank.

### 3.3. Pixel-Level Defect Detection

After completing the construction of memory bank, the detection process involves several steps as illustrated in Figure 3. First, the test image is fed into the corresponding feature extractor, which is then used to extract multi-scale intermediate features of the image. Next, the obtained intermediate features are then compared to the feature prototypes obtained during the model construction stage, and based on their similarity, corresponding similarity feature maps are calculated.

The features obtained from the test images are compared to the corresponding multi-scale normal and defect prototypes at each position using a similarity calculation s(·). The similarity calculation between input and prototypes is shown in Equations (6) and (7). $S_{D}^{k}(x)(h,w)$ refers to the similarity between defect feature prototypes and input image feature map at position (*h*,*w*), similarly $S_{N}^{k}\left(x\right)\left(h,w\right)$ refers to the similarity between normal feature prototypes and input image feature map. Specifically, $f_{img}^{k}$ denotes feature map of a input image.
(6)$$S_{D}^{k}\left(x\right)(h,w)=\frac{1}{n}\sum_{i=1}^{n}s\left(p_{D}^{k}\left(x_{i}\right),f_{img}^{k}\left(h,w\right)\right),S_{D}^{k}(x)(h,w)\in R$$
(7)$$S_{N}^{k}\left(x\right)\left(h,w\right)=\frac{1}{n}\sum_{i=1}^{n}s\left(p_{N}^{k}\left(x_{i}\right),f_{img}^{k}\left(h,w\right)\right),S_{N}^{k}(x)(h,w)\in R$$

In this study, cosine similarity was chosen for similarity calculation. The calculation process for the similarity feature map is demonstrated in Equation (8), where the defect prototype $p_{D}^{k}(x_{j})$ and input image feature map $f_{img}^{k}(h,w)$ are both vectors of length $C_{k}$.
(8)$$s\left(p_{D}^{k}\left(x_{j}\right),f_{img}^{k}\left(h,w\right)\right)=\frac{p_{D}^{k}(x_{j})\cdot f_{img}^{k}(h,w)}{{\left\|p_{D}^{k}(x_{j})\right\|}_{2}\times {\left\|f_{img}^{k}(h,w)\right\|}_{2}}$$

After performing similarity calculations between all feature prototypes and the input image features, a probability distribution over defects for each position in the image is established using softmax. This allows us to obtain the probability of each position being a defect, as shown in Equation (9), where $q(y=defect\left|x\right.)$ represents the conditional probability that y belongs to defect under the premise of given input *x*:(9)$$q(y=defect\left|x)=\right.\frac{1}{3}\sum_{k=1}^{3}\frac{exp(S_{D}^{k}(x)(h,w)}{exp(S_{D}^{k}\left(x\right)\left(h,w\right)+exp(S_{N}^{k}(x)(h,w))}$$

By combining Equation (9), we can observe that the essence of FS-RSDD is to evaluate the similarity between input samples and defect prototypes, as well as the dissimilarity between input samples and normal prototypes in three feature spaces (obtained from three layers of the feature extractor), as illustrated in Figure 4. Finally, defect detection is performed by integrating the prediction results from the three feature spaces, as shown in Equation (9).

### 3.4. Image-Level Defect Detection

Image-level defect detection aims to perform image-level binary classification between normal rail images and rail images containing defects. By processing the predicted results in Section 3.3 accordingly, we can obtain the corresponding image-level prediction results.

Our approach is based on a simple idea. If we define $q(y=defect\left|x)\right.$ in Section 3.3 as the defect score of a certain pixel, we can represent the probability of an image containing defects by considering the defect score of the pixel with the highest defect score in the predicted image. However, this approach leads to poor performance, as it only considers individual pixels and lacks consideration for the local neighborhood pixels. In order to further improve the detection accuracy, we decided to use a simple Gaussian blur to fuse information from the local neighborhood of pixels. The process of Gaussian blur on an image is the convolution of the image with a two-dimensional Gaussian distribution that has been discretely sampled, as shown in Figure 5. Subsequently, we performed image-level defect detection. This approach significantly improved the performance of our model, as demonstrated in Section 4.3.

## 4. Experiments and Results

### 4.1. Evaluation Metrics

This article focuses on the detection and localization of rail surface defects, which involves binary classification tasks at both image and pixel levels for defect rail images and normal rail images. The receiver operating characteristic (ROC) and precision recall (PR) are used as the evaluation metrics for the model.

These two performance metrics have different emphases, which enable this study to comprehensively evaluate the performance of the model during the experimental process. Additionally, we assessed the classification performance at both the image level and the pixel level. These two metrics, respectively, represent the algorithm’s ability to classify defects and accurately locate them. By evaluating performance at both levels, a more comprehensive analysis of the algorithm’s effectiveness can be obtained.

As defined in Equations (10) and (11), the *x*-axis of the ROC curve represents the false-positive rate (*FPR*), and the *y*-axis represents the true-positive rate (*TPR*), in which *FP* denotes false positives (negative samples falsely predicted as positive), *TN* denotes true negatives (negative samples correctly predicted as negative), *TP* denotes true positives (positive samples correctly predicted as positive), and *FN* denotes false negatives (positive samples falsely predicted as negative). A larger area under the ROC curve indicates better performance of the classifier. In this article, the model evaluation metrics are divided into image-level ROCs and pixel-level ROCs, which correspond to evaluation metrics for images and individual pixels, respectively.
(10)$$FPR=\frac{FP}{FP+TN}$$
(11)$$TPR=\frac{TP}{TP+FN}$$

The *recall* rate is represented on the *x*-axis of the *PR* curve, while the accuracy precision is depicted on the vertical axis. The definitions of recall and *precision* are provided in Equations (12) and (13), respectively. The area under the *PR* curve corresponds to the average accuracy (AP). A larger area under the *PR* curve indicates better performance of the classifier.
(12)$$Recall=\frac{TP}{TP+FN}$$
(13)$$Precision=\frac{TP}{TP+FP}$$

### 4.2. Experiment Setup

#### 4.2.1. Dataset Setup

This article uses a dataset from the open-source Rail Surface Defect Detection dataset (RSDDS) [45]. RSDDS consists of two types of rail defect data: Type-I and Type-II. Type-I defects were obtained from 67 defect images collected from high-speed train tracks. Type-II defects, on the other hand, were collected from 128 defect images obtained from regular/heavy-duty transportation tracks. In this article, the two types of defect images are first divided into normal samples and defect samples through fixed ratio image cropping. During the cropping process, images that have a too small defect area are discarded. The image processing process is shown in Figure 6.

After the aforementioned process, there are 113 defect samples in Type-I dataset and 230 defect samples in Type-II dataset. To ensure a balanced representation of positive and negative samples in the test set and to provide a more accurate evaluation of the performance of the proposed method, we randomly selected normal rail samples for the test set, ensuring that the quantity was consistent with the number of defect samples.

Finally, the Type-I dataset consisted of 302 normal samples for the training set, 113 defect samples, and 113 normal samples for the testing set. Meanwhile, the Type-II dataset comprised 2071 normal samples for the training set, 230 defect samples, and 230 normal samples for the testing set. Additionally, the model necessitates a limited number of defect samples during the training phase, which will be randomly selected from the test set. After being partitioned and resized, the resolution of Type-I rail images is 160 × 160, while Type-II rail images have a resolution of 64 × 64.

#### 4.2.2. Comparison Experiment Setup

The proposed method in this article is compared with mainstream unsupervised industrial defect detection algorithms and existing few-shot supervised industrial defect detection algorithms in terms of classification evaluation metrics on the RSDDS dataset.

As there may be variations in the defect samples extracted during each training process, a random selection of a small subset of defect samples is employed for training during the experimental process. To ensure robustness, multiple experiments are conducted, and the average value is considered as the validation result of the model.

In the comparison experiments with unsupervised methods, since the defect samples for training are randomly selected in each experiment, the test set may not include the exact same defect samples in each experiment. Therefore, to maintain consistency, multiple tests are also conducted on the unsupervised industrial defect detection algorithms, and the defect samples utilized for our method are excluded from the test set to ensure a fair evaluation of both methods on the same test set, ensuring that the test set used aligns consistently with the test set employed in each experiment of the proposed method in this article.

Similarly, when comparing the performance with few-shot supervised industrial defect detection algorithms, multiple experiments are conducted, and the average test results are used as the final performance metric. Additionally, in each experiment, the defect samples utilized for training the few-shot supervised industrial defect detection algorithm are consistent with the defect samples randomly selected for training in the proposed method in this article. Furthermore, the default values were maintained for all other settings of the comparative models in the code. All the comparative models that were involved with the gradient decent process are trained to convergence to guarantee the impartiality of performance comparisons.

### 4.3. Comparison with Unsupervised-Based Algorithm

The performance comparison results with unsupervised methods are presented in Table 1, Table 2 and Table 3. Table 1 displays the average image-level ROC, Table 2 shows the average image-level AP, and Table 3 presents the average pixel-level ROC. All of these metrics were obtained from 20 random sampling validations. In the training process, “m” refers to the defect sample used. It is worth mentioning that the unsupervised algorithms were implemented using the open-source industrial defect detection library anomalib [46].

Combining the data from Table 1 and Table 2, it can be observed that our proposed method outperforms other unsupervised industrial defect detection algorithms in terms of image-level classification ROC, except for PatchCore. However, it does not show significant advantage over other unsupervised algorithms in terms of image-level AP.

The reason behind this result lies in the fact that AP is more inclined towards the detection of positive instances, i.e., defect samples, while ROC is a relatively balanced evaluation metric. The better performance of our method in ROC compared to AP may be attributed to the fact that, while maintaining a high precision, our method has a lower false-positive rate for defect detection. However, it has a higher false-negative rate compared to some algorithms, while under the same conditions, some unsupervised defect detection algorithms have a lower false-negative rate but a higher false-positive rate.

We further analyzed the defects that were not successfully detected by our method. Figure 7 shows the heatmap of the undetected defect samples and the successfully classified normal samples by our method, under the given defect detection threshold.

By observing the heatmap of false-negative samples, we can visually see that the defective parts in the rail images are actually represented by darker colors. This means that our proposed method can accurately distinguish the defect foreground from the normal rail background. The reason why these defects were not detected can be further observed from the predicted results of true-negative samples. We can see that the reason for the lower AP in our method is that for those stains or noises that are difficult to distinguish from defects in the images, our method also considers them as potential defects. Although the probability of these noises belonging to defects may not be significantly higher than true defects, this ambiguous discrimination leads to our method’s inability to provide clear judgments for some challenging cases. In other words, the trade-off of our method rarely misclassifying normal samples as defect samples is that some defect samples are also considered as normal samples. As a result, we have a higher ROC but a relatively lower AP.

Another thing we can observe from Table 1 and Table 2 is that, regardless of the algorithm used, there is a significantly better performance on Type-II data compared to Type-I data. The reason behind this phenomenon is consistent with our previous analysis on the difference in performance between the two metrics, which is the presence of noise and interference in the images. As shown in Figure 8, it can be seen that, perhaps due to better image acquisition conditions, the Type-II rail images contain much less noise compared to Type-I data.

In Figure 8, the red curve indicates the defective area, while the green curve indicates the noise that is similar to the defect. It can be clearly seen that Type-I data contain much more noise that interferes with defect detection compared to Type-II data.

Furthermore, according to the data in Table 3, we can also observe that our method outperforms most unsupervised AD methods except Patchcore in terms of pixel-level ROC, indicating that our algorithm achieves more precise segmentation for the same defect.

In Figure 9, the segmentation results of different algorithms for the same defect sample are displayed. It is evident from this that our proposed algorithm exhibits more precise prediction and is less prone to generating false predictions on the background when compared to other algorithms.

### 4.4. Comparison with Few-Shot Supervised-Based AD Algorithms

We also conducted comparative experiments with few-shot industrial defect detection algorithms. The experimental setting was similar to the unsupervised algorithm comparison experiment. We conducted 20 experiments, each time randomly selecting m defect samples for model training. The defect samples used for training in the comparative methods remained consistent with our proposed method. The average ROC and average AP were then calculated for performance comparison, as shown in Table 4. According to the results in the table, considering both the ROC and AP metrics, our method demonstrates advantages compared to DevNet [23] and DRA [22].

We not only compared the average performance but also recorded the performance of the model for each experiment in order to observe the impact of different training samples on the model’s performance.

Figure 10 shows the changes in the model’s ROC after training with randomly sampled defect data from Type-I and Type-II datasets, respectively.

From the observation of Figure 10, it can be noticed that FS-RSDD exhibits more stable and robust performance compared to other models when different defect data are used for training. Furthermore, although FS-RSDD shows more fluctuations on Type-I data compared to Type-II data, it shows better performance compared to the rest of the few-shot supervised-based AD models. This is mainly attributed to the fact that FS-RSDD not only utilizes defect features but also fully utilizes the features of a normal rail for defect detection. On the other hand, other algorithms tend to focus more on extracting information from defect samples, which can lead to lower accuracy when the quality of defect samples is poor.

### 4.5. Ablative Studies

In this section, we conducted ablation experiments to explore the impact of different settings on the performance of FS-RSDD. These experiments included comparative experiments on the model’s performance using features from different semantic levels of the feature extractor, whether using Gaussian blur or not, and extracting features using different feature extractors. The experiments were conducted by extracting defect samples and training the model 20 times and then comparing the average performance of the model. The number of samples extracted was m = 10, and the defect samples used for training were consistent with those used in the experiments in Section 4.1 and Section 4.2.

We first conducted comparative experiments on the performance of the FS-RSDD model using different feature extractors, both with and without Gaussian blur. Table 5 presents the performance of FS-RSDD on Type-I and Type-II rail surface defect datasets when using ResNet18, ResNet50, and WideResNet50 [47] as feature extractors.

From the experiment data in Table 5, we can clearly observe the significant impact of different feature extractors and the use of Gaussian blur on the detection performance of the model. From this, we can observe that when using ResNet18 as the feature extractor, the model has lower accuracy but faster speed. This is evident due to ResNet18 having fewer model parameters and faster inference speed but correspondingly poorer feature extraction capability. On the other hand, unlike ResNet18, WideResNet50, with its wider feature channels, can achieve better performance when used as a feature extractor, albeit with relatively slower detection speed.

Additionally, by comparing the performance of each feature extractor with and without Gaussian blur, we can easily observe the extent of improvement that Gaussian blur brings to the model’s performance. This demonstrates the enhancement of predictive performance through the fusion of pixel neighborhood features.

Comparative experiments were also conducted on the performance of the FS-RSDD model by utilizing features from different semantic levels to construct the memory bank, as presented in Table 6. In the experiment, the WideResNet50 is employed as the feature extractor, and Gaussian blur is applied to improve the performance. It can be clearly seen that the use of different combinations of semantic level features has an impact on the performance of FS-RSDD. When only using single- or two-level features for model construction, the performance of the model is suboptimal. However, when using multi-level features from a shallow layer, FS-RSDD exhibits the best performance. Furthermore, we conducted experiments with the utilization of features from deeper semantic levels. However, we observed no significant enhancement in the performance of FS-RSDD but a decrease in frames per second (FPS) due to the increased number of feature similarity calculations.

### 4.6. Time Complexity and the Size of Memory Bank

In this section, we conducted a comparative analysis of the computational complexity and size of memory banks among different models. The Type-I image data have a resolution of 160 × 160, while the Type-II data have a resolution of 64 × 64. All models employed the same network, WideResNet50, as the feature extractor. It is evident from Table 7 that FS-RSDD, benefiting from its compact feature memory bank that models the entire normal rail background, outperforms unsupervised anomaly detection and few-shot defect detection algorithms in terms of time complexity. Moreover, this advantage is more significant on the low-resolution Type-II dataset.

We also conducted experiments regarding the size of the memory bank. We extracted the memory bank of our model and other methods and compared them to demonstrate the compactness of the memory bank obtained by our approach. We contrasted our method with memory-bank-based approaches [30,32]. The results are as shown in Table 8. In the experiment, each model utilizes the WideResNet50 feature extractor, with an input image resolution of 160.

## 5. Discussion

The method proposed in this article mainly combines the idea of feature-embedding-based industrial defect detection algorithms and the prototypical network. By embedding features of defects and normal rails, corresponding feature memory banks are obtained. FS-RSDD estimates the similarity of the input samples to the defect prototype and the normal prototype in the feature space for defect detection. This simple and direct method can achieve quite good results on the rail surface defect dataset using only a few samples. However, there are still some shortcomings. After studying the experimental results in Section 4.3, it can be concluded that although this method can effectively distinguish the rail background and defect foreground, it cannot effectively discriminate between defects and noise.

To explore the possibility of improving the model’s detection performance using traditional image-processing techniques, we conducted additional experiments. We performed another experiment on both the RSDDS Type-I dataset without processing and the dataset processed with image processing. The experiment was conducted only once, and the random seed was fixed. Following the method described in reference [45], we used gamma transform to improve the uneven lighting in the images and combined it with Gaussian blur for image denoising. In the end, we achieved a pixel-level ROC of 99.3% and an image-level ROC of 96.5% on the original dataset, while on the denoised dataset we achieved a pixel-level ROC of 99.3% and an image-level ROC of 96.2%. It can be seen that after image processing, the model’s performance did not improve as expected. We believe this may be due to the fact that deep-learning-based feature extractors have strong feature extraction capabilities, and the noise and uneven lighting that traditional image processing techniques can handle can also be distinguished by the feature extractor. However, noise and interference that are difficult for the feature extractor to distinguish are equally challenging for traditional image processing techniques. Therefore, instead of traditional image processing methods, we will focus on enhancing our work through novel image processing techniques in the future. Additionally, our research will prioritize exploring deep learning methods in defect detection.

## 6. Conclusions

This paper proposes a few-shot rail surface defect detection model, FS-RSDD, to address the issue of insufficient defect samples in the field of rail surface defect detection. FS-RSDD combines the idea of feature-embedding-based industrial defect detection algorithms with the prototypical network. The method utilizes a pre-trained convolutional neural network to embed features of both defective and normal samples. It then uses clustering algorithms to learn the distribution of features of normal samples. Finally, through the prototype learning approach, softmax is used to estimate the probability of a test sample’s feature belonging to a defect in the feature space.

The proposed method surpasses all comparative algorithms in terms of speed by achieving a compact feature memory bank, which models the overall feature distribution of normal rail backgrounds. Additionally, the proposed method outperforms comparative few-shot defect detection algorithms in terms of accuracy on the RSDDS public dataset and is on par with the current state-of-the-art unsupervised anomaly detection algorithms.

## Figures and Tables

**Figure 1 sensors-23-07894-f001:**
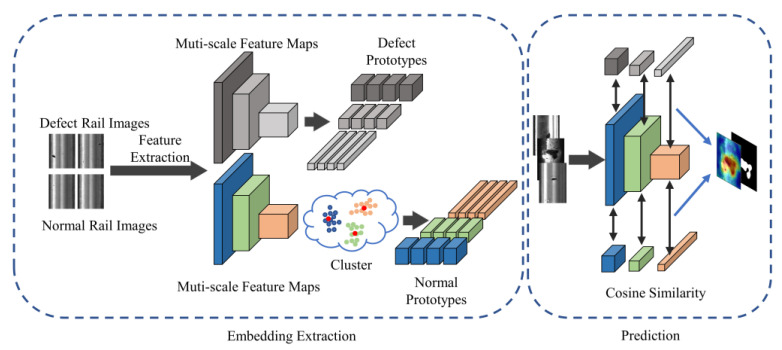
The architecture of proposed model.

**Figure 2 sensors-23-07894-f002:**
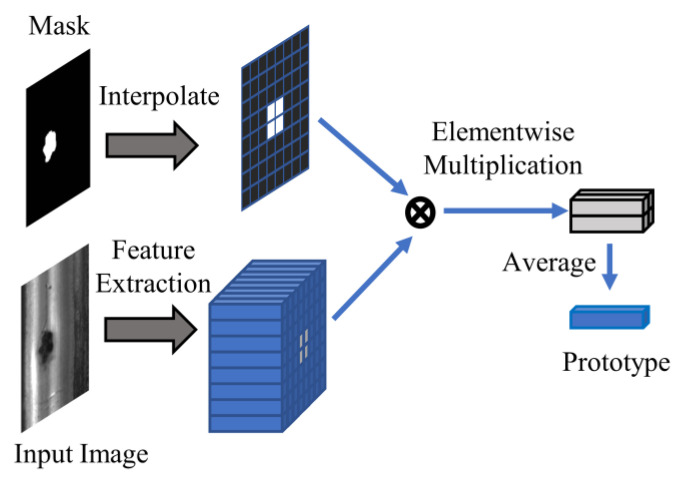
Mask average pooling.

**Figure 3 sensors-23-07894-f003:**
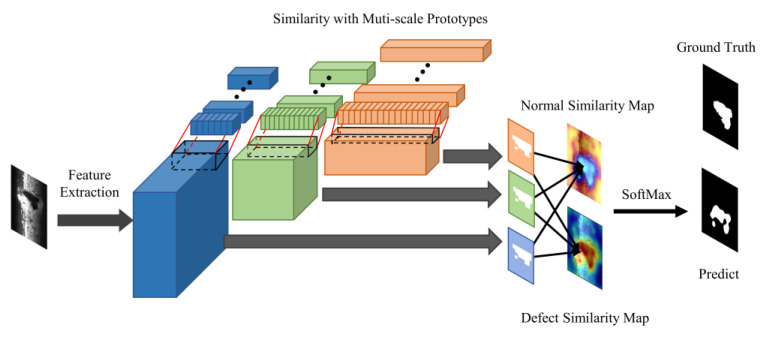
Detection procedure of FS-RSDD.

**Figure 4 sensors-23-07894-f004:**
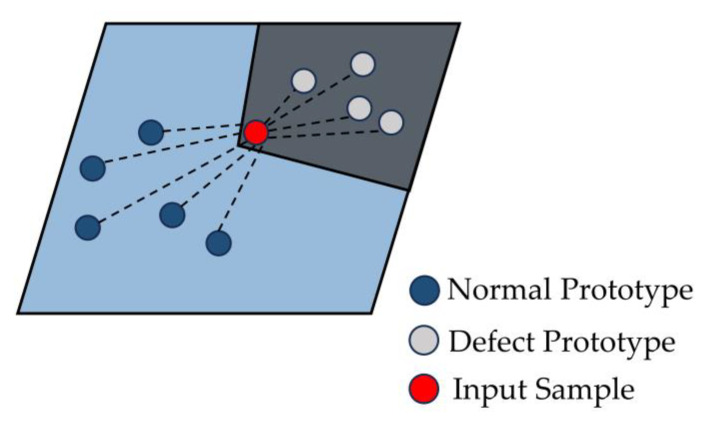
The similarity calculation of the FS-RSDD, blue and black areas represent the distribution bound of normal and defective samples, respectively.

**Figure 5 sensors-23-07894-f005:**
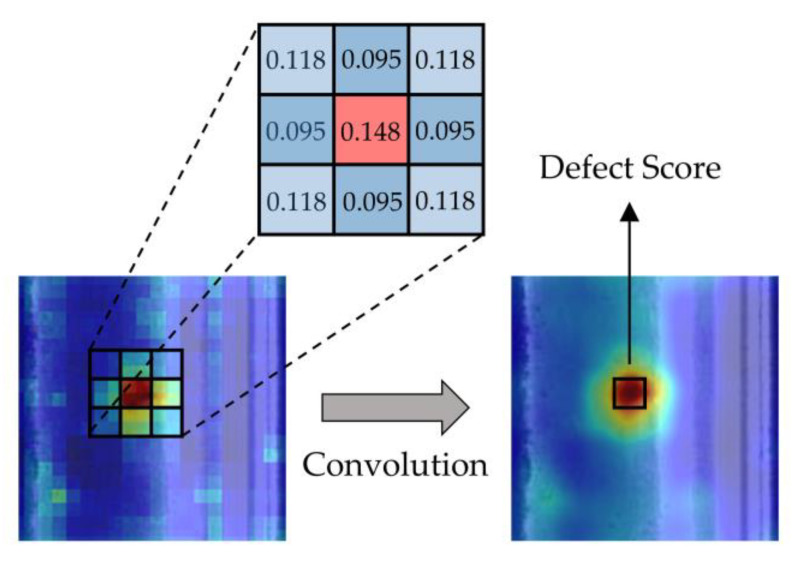
The schematic of the process of two-dimensional Gaussian blur, in heat map, the depth of red color represents the probability of the presence of defects in the area.

**Figure 6 sensors-23-07894-f006:**
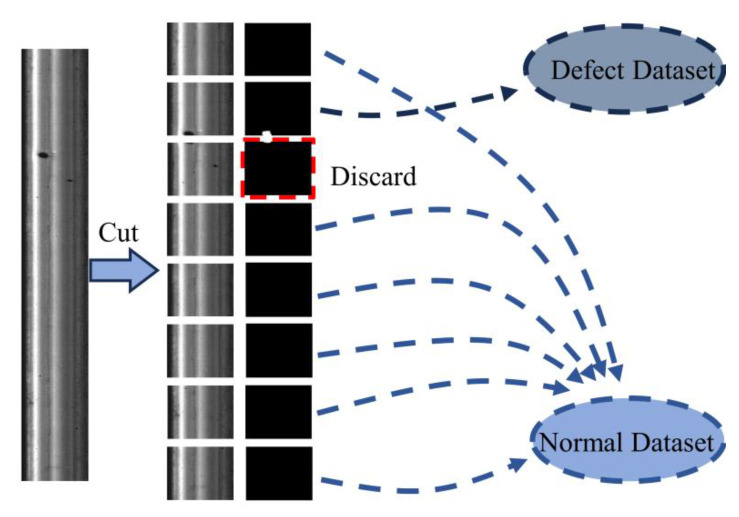
The process of dataset creation.

**Figure 7 sensors-23-07894-f007:**
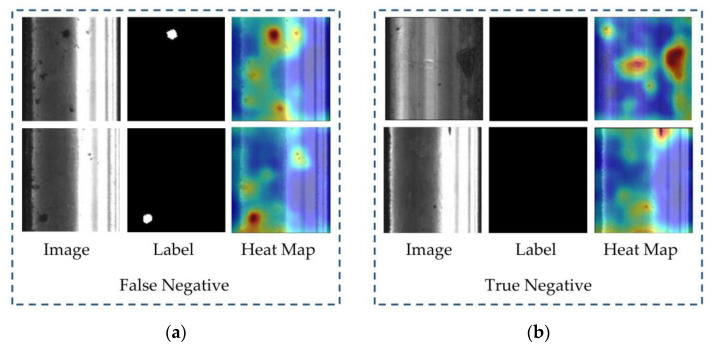
The heatmap visualization of false negatives and true negatives in the prediction results: (**a**) heatmap of false negatives, showing high defect scores in the actual defective regions; (**b**) heatmap of true negatives, showing high anomaly scores for noise or stains that are similar to defects.

**Figure 8 sensors-23-07894-f008:**
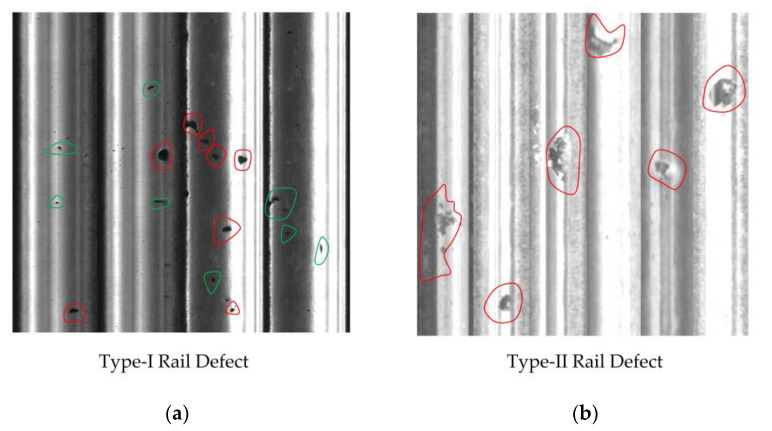
Type-I rail surface data and Type-II rail surface data. The true defects are circled in red, while the noise or stains similar to defects are circled in green. (**a**) Type-I data, where more noise and stains are visible; (**b**) for Type-II data, it is visually evident that there is not much noise interference.

**Figure 9 sensors-23-07894-f009:**
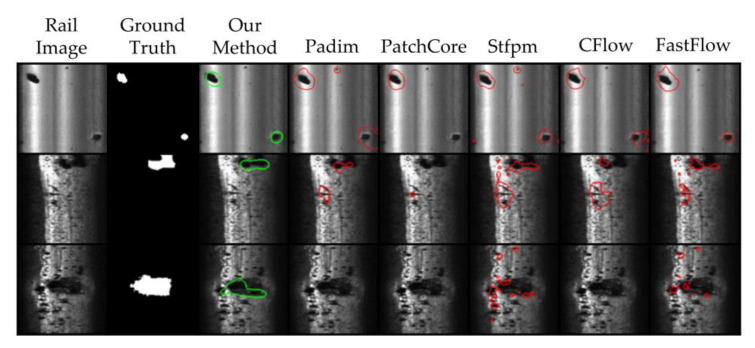
Comparison of FS-RSDD with other unsupervised AD models in terms of prediction results.

**Figure 10 sensors-23-07894-f010:**
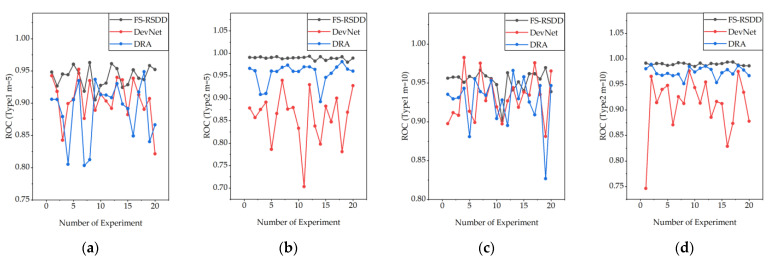
Performance fluctuations of FS-RSDD compared to the benchmarked few-shot supervised-based AD algorithms with different defect data for training: (**a**) Type-I, m = 5; (**b**) Type-II, m = 5; (**c**) Type-I, m = 10; (**d**) Type-II, m = 10.

**Table 1 sensors-23-07894-t001:** Image-level ROC of our proposed FS-RSDD and other unsupervised anomaly detection models.

	Model					
Dataset	FS-RSDD	Padim [30]	PatchCore [32]	stfpm [31]	cflow [9]	fastclow [10]
RSDDS Type-I m = 5	0.941	0.950	0.951	0.899	0.866	0.895
RSDDS Type-I m = 10	0.952	0.950	0.951	0.905	0.903	0.894
RSDDS Type-II m = 5	0.989	0.976	0.996	0.990	0.875	0.983
RSDDS Type-II m = 10	0.991	0.976	0.996	0.990	0.881	0.984

**Table 2 sensors-23-07894-t002:** Image-level AP of our proposed FS-RSDD and other unsupervised anomaly detection models.

	Model					
Dataset	FS-RSDD	Padim	PatchCore	stfpm	cflow	fastclow
RSDDS Type-I m = 5	0.943	0.981	0.982	0.947	0.939	0.956
RSDDS Type-I m = 10	0.953	0.980	0.981	0.951	0.950	0.953
RSDDS Type-II m = 5	0.986	0.970	0.995	0.988	0.890	0.980
RSDDS Type-II m = 10	0.986	0.970	0.995	0.988	0.886	0.981

**Table 3 sensors-23-07894-t003:** Pixel-level ROC of our proposed FS-RSDD and other unsupervised anomaly detection models.

	Model					
Dataset	FS-RSDD	Padim	PatchCore	stfpm	cflow	fastclow
RSDDS Type-I m = 5	0.987	0.976	0.974	0.980	0.970	0.953
RSDDS Type-I m = 10	0.991	0.977	0.975	0.981	0.971	0.954
RSDDS Type-II m = 5	0.961	0.920	0.919	0.948	0.852	0.919
RSDDS Type-II m = 10	0.962	0.920	0.920	0.948	0.855	0.919

**Table 4 sensors-23-07894-t004:** Comparison between FS-RSDD and other few-shot industrial defect detection models.

	ROC			AP		
Dataset	FS-RSDD	DevNet	DRA	FS-RSDD	DevNet	DRA
Type-I m = 5	0.941	0.905	0.888	0.943	0.967	0.958
Type-I m = 10	0.952	0.930	0.927	0.953	0.976	0.973
Type-II m = 5	0.989	0.858	0.955	0.986	0.901	0.963
Type-II m = 10	0.991	0.911	0.974	0.986	0.939	0.978

**Table 5 sensors-23-07894-t005:** The impact of different feature extractors and the use of Gaussian blur on the performance of FS-RSDD.

Feature Extractor		Image-Level ROC	Image-Level AP	Pixel-Level ROC	FPS
Type-I m = 10	ResNet18	0.896	0.889	0.964	130.890
	ResNet50	0.906	0.901	0.976	70.403
	WideResNet50	0.930	0.927	0.985	64.664
	ResNet18 + Gaussian blur	0.934	0.935	0.980	130.052
	ResNet50 + Gaussian blur	0.937	0.935	0.986	70.837
	WideResNet50 + Gaussian blur	0.952	0.953	0.991	63.519
Type-II m = 10	ResNet18	0.968	0.959	0.939	299.114
	ResNet50	0.984	0.978	0.948	231.154
	WideResNet50	0.985	0.979	0.952	204.306
	ResNet18 + Gaussian blur	0.986	0.983	0.955	265.375
	ResNet50 + Gaussian blur	0.990	0.986	0.959	215.554
	WideResNet50 + Gaussian blur	0.990	0.987	0.959	211.447

**Table 6 sensors-23-07894-t006:** The impact of constructing a feature memory bank using different hierarchical features on the performance of FS-RSDD.

Dataset	Layer	Image-Level ROC	Pixel-Level ROC	Image-Level AP	FPS
Type-I m = 10	Layer1	0.677	0.885	0.658	71.145
	Layer1 + 2	0.920	0.976	0.911	63.552
	Layer1 + 3	0.935	0.988	0.943	66.293
	Layer2 + 3	0.954	0.991	0.956	75.039
	Layer1 + 2 + 3	0.952	0.991	0.953	63.519
	Layer1 + 2 + 3 + 4	0.949	0.988	0.948	61.557
Type-II m = 10	Layer1	0.846	0.925	0.868	235.983
	Layer1 + 2	0.981	0.964	0.976	221.524
	Layer1 + 3	0.990	0.957	0.986	222.655
	Layer2 + 3	0.990	0.959	0.987	240.008
	Layer1 + 2 + 3	0.991	0.962	0.987	211.447
	Layer1 + 2 + 3 + 4	0.992	0.955	0.988	207.422

**Table 7 sensors-23-07894-t007:** The time complexity comparison between FS-RSDD and other benchmarked models.

	FS-RSDD	Padim	Patchcore	stfpm	cflow	fastflow	devnet
Type-I m = 10	63.519	57.434	60.378	47.500	21.310	44.954	25.306
Type-II m = 10	211.447	101.045	96.905	84.843	61.526	58.613	39.192

**Table 8 sensors-23-07894-t008:** Memory bank comparison: each element is a floating-point number.

	FS-RSDD	Patchcore	Padim
Number of elements	71,680	184,320	950,364,800
File size	282 KB	721 KB	3.54 GB

## Data Availability

Not applicable.

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
