# Peer review of "FS-RSDD: Few-Shot Rail Surface Defect Detection with Prototype Learning"

_sensors, 2023, doi:10.3390/s23187894_

Round 1

Reviewer 1 Report

This paper proposed a model called FS-RSDD (Few-shot Rail Surface Defect Detection) to detect the rail surface defect by few samples. This model has two steps, a pre-trained model to extract deep features and an unsupervised learning method to learn feature distribution.  The performance of the proposed model is validated by experiment on a dataset. There are a few major comments as follows.

1.  In section 1 introduction, the motivation of proposing the new deep learning method to detect the rail surface defects is provided. While are the non-deep learning methods like image processing algorithm able to detect the rail surface defects? Because deep learning methods require more image samples and their labels, and the focus on this paper is on surface defect identification. If look at the data samples in Figure.9, some image processing algorithms may be able to detect the defects by simply identify the masks.

2.   In Figure 2, the mask average pooling, what’s the resolution (size) of the Mask? And what’s the size of features? Do the mask and feature have the same resolution (size)? And how is the ground truth mask obtained?

3.   In section 3.3 pixel-level defect detection, is the mask of testing image needed for detection process?

4.   In section 4.2.2 comparison experiment setup, the proposed method is compared with mainstream unsupervised algorithms and existing few-shot supervised industrial defect detection algorithms. While what are the settings on training different models? What the training convergence of different methods?

5.   In section 4.4, the comparison with supervised learning algorithm, the comparison was only conducted on DevNet and DRA. While what the metrics look like if use the simple object detection networks, like YOLO? Because one purpose of this paper is to tackle the issue of limited data for object detection. It would be convincing to show the outperformance of the proposed method to existing object detection networks.

6.   From pixel property point of view, the stains or noises show the same property as defects on pixels, which means it may not be possible to differentiate the stains or noise and real defects. Is it possible to solve this challenge, for example enhancing the images?

Author Response

We sincerely thank you for your professional and helpful review work. Please look at the attachment, which includes responses to all the questions and instructions for manuscript modifications. Thank you again for your review of our work.

Reviewer 2 Report

The paper deals with the problem of detecting surface defects on rails by analyzing their images. For this purpose the authors consider the application of machine learning methods. In the manuscript the authors propose the application of a few-shot rail surface defect detection model based on unsupervised learning. The effectiveness of the proposed approach is verified by the comparative study with another unsupervised anomaly detection models. 

The remarks are as follows:

1. It is worth mentioning the tools used for the development of the model (i.e. libraries used).

2. Some letters are missing in the title of Figure 1.

3. The headings of figures 4 and 5 are too long and overloaded with data. Please consider shortening them.

Author Response

(The authors gave the same response as above.)

Reviewer 3 Report

This paper proposed a few-shot rail surface defect detection model, FS-RSDD. But the format and expression of this manuscript need further improvement. And the explanation of proposed method is insufficient. I think this manuscript should be major revisions.

Main problems:

1.     This manuscript lacks a basic introduction of the related work on few-shot learning.

2.     This paper emphasizes building a compact multi-scale memory bank, but after reading the whole manuscript, l cannot find any description or experimental comparison of “compact”.

3.     The way to obtain samples by a fixed ratio image cropping may lead to changes in the original characteristics of some defects, which in turn affects the modeling.

4.     The author’s description of Figure1 is proposed method consists of modeling and detection, but Figure1 shows proposed method consists of embedding extraction and prediction. Please ensure that the figure is consistent with the description of it.

5.     Line 45: The subject and verb do not agree in number. Please ensure that they match in singular/plural form.

6.     Line 80 & Line 86: Please review the punctuation usage for accuracy and consistency.

7.     Line 218: ”muti-scale” should be “multi-scale”

Too many typos

Author Response

(The authors gave the same response as above.)

Round 2

Reviewer 1 Report

All the comments are tackled in the response letter.